# Towards Circulating-Tumor DNA-Based Precision Medicine

**DOI:** 10.3390/jcm8091365

**Published:** 2019-09-02

**Authors:** Ai Hironaka-Mitsuhashi, Anna Sanchez Calle, Takahiro Ochiya, Shin Takayama, Akihiko Suto

**Affiliations:** 1Division of Molecular and Cellular Medicine, National Cancer Centre Research Institute, 5-1-1 Tsukiji, Chuo-ku, Tokyo 104-0045, Japan; 2Department of Molecular and Cellular Medicine, Institute of Medical Science, Tokyo Medical University, 6-7-1 Nishishinjuku, Shinjuku-ku, Tokyo 160-0023, Japan; 3Department of Breast Surgery, National Cancer Centre Hospital, 5-1-1 Tsukiji, Chuo-ku, Tokyo 104-0045, Japan

**Keywords:** biomarker, breast cancer, circulating-tumor DNA, colorectal cancer, lung cancer, precision medicine, targeted therapy

## Abstract

In the era of precision medicine, targeted therapies have been implemented for various diseases. Genomic information guides decision-making in cancer treatment. The improvements in next-generation sequencing and polymerase chain reaction have made it possible to access the genetic information using circulating-tumor DNAs (ctDNAs). Molecular characteristics of individual tumors can be obtained by analysis of ctDNAs, thus making them excellent tools to guide decision-making during treatment. In oncology, the use of ctDNAs in clinical practice is now gaining importance. Molecular analysis of ctDNAs has potential for multiple clinical applications, including early diagnosis, prognosis of disease, prognostic and/or predictive biomarkers, and monitoring response to therapy and clonal evolution. In this paper, we highlight the applications of ctDNAs in cancer management, especially in metastatic setting, and summarize recent studies about the use of ctDNAs as predictive biomarkers for the therapeutic adaptation/response in lung cancer, breast cancer, and colorectal cancer. These studies offer the evidence to use ctDNAs as a promising approach to solve unmet clinical needs.

## 1. Introduction

Precision medicine is an emerging approach for both prevention and treatment of disease [1,2,3]. It is important to develop more precise diagnostic tools in precision medicine for a variety of diseases including cancer. In oncology, precision medicine aims to select effective treatment based on the molecular characteristics of individual tumors. Genomic information is important for treatment strategy, since targeted therapy is directed against key signaling pathways involved in cancer growth and malignant progression [4,5,6,7,8,9,10,11,12,13,14]. Recently, improvements in the next-generation sequencing (NGS) and polymerase chain reaction (PCR)-based approaches have enabled the identification of genomic information by using biomarkers such as circulating tumor cells, cell-free DNAs (cfDNAs), circulating RNA including mRNA and secreted microRNAs in peripheral blood, and other body fluids, collectively termed liquid biopsies [15,16,17,18,19,20]. Applications of liquid biopsy include early diagnosis, prognosis of disease, prognostic and/or predictive biomarkers for the therapeutic adaptation/response, and monitoring response to therapy and clonal evolution.

Among liquid biopsies, cfDNAs are of particular interest [19,21,22]. The first publication on the presence of cell-free nucleic acid in the blood of healthy individuals was attributed to Mandel et al. in 1948 [23]. The next impressive study was the report published in 1977 that demonstrated increasing levels of cfDNAs in the blood of cancer patients in comparison with healthy individuals [24]. Molecular analysis of cfDNAs has been actively researched in cancer management since previous studies confirmed that elevated levels of cfDNAs were detected in various kinds of solid tumors regardless of tumor progression [25]. As cancer progresses, the levels of cfDNAs increase and cfDNAs from tumor cells are likely to have genomic alterations corresponding to the tumor tissues [25,26,27,28,29,30,31,32,33,34,35,36]. Since cfDNAs can be an alternative to the tumor tissue, their utility has been exploited in the management of cancer, especially for lung cancer, breast cancer, and colorectal cancer (Figure 1).

On the other hand, the actual mechanism of secretion of cfDNAs has not yet been elucidated [17]. At present, cfDNAs are thought to be released from cells, mostly through apoptosis and necrosis, and possibly also by active secretion [19,37]. In this review, we use the term circulating-tumor DNAs (ctDNAs) instead of cfDNAs since we focused on circulating DNA fragments directly derived from tumor cells. In oncology, the use of ctDNAs in clinical practice for diagnosis/prognosis is gaining importance. For instance, EGFR (epidermal growth factor receptor) mutation testing using ctDNAs was approved as a companion in vitro diagnostic, while ctDNA testing has been required for the pre-analytical and analytical phase in the other cancer [38,39,40]. The approval of molecular analysis of cfDNAs should open the door for the approval of other tests, not only for the prediction of therapeutic responses, but also to monitor tumor burden [21]. In our review, we will highlight the applications of cfDNAs in cancer management (Figure 2), especially in a metastatic setting, and summarize recent studies about use of cfDNAs as promising biomarkers for lung cancer, breast cancer, and colorectal cancer. These studies offer evidence in favor of cfDNAs to be used as reliable tools to solve unmet clinical needs.

## 2. ctDNAs Are Promising Biomarkers in Oncology

ctDNAs can be advantageous among liquid biopsies owing to improvements in sensitivity of the techniques to capture ctDNAs. The ctDNAs are at low levels in cfDNAs, and thus highly sensitive technologies are required for their detection [17,27,41]. The presence of specific mutations in cancer helps to distinguish ctDNAs from normal cfDNAs. At present, digital-PCR has enabled the detection of rare mutations in cfDNAs with allele fractions as low as 0.001% in a wild-type background [42]. NGS provides simultaneous characterization of somatic mutations such as single-nucleotide variants, insertions/deletions, structural rearrangements, and copy-number alterations. Aside from mutational alteration, epigenetic alterations such as methylation of promoter/enhancer can be measured by using ctDNAs. The increasing availability and reliability of these techniques has been facilitating novel, high-sensitivity applications for ctDNAs [19]. It has been reported that molecular analysis of ctDNAs can guide treatment decision, however, ctDNA concentration values are biased in the literature, since in most cases, ctDNAs are determined from mutations of a few or a panel of genes [12,16,17,19,21,26,29,43,44,45].

Genomic information decides adaptations for targeted therapy. Examples of targeted therapies include amplification of human epidermal growth factor receptor2 (HER2) for HER2 antibody in breast cancer and in gastric cancer and activating EGFR mutations for EGFR tyrosine kinase inhibitors (TKIs) in non-small cell lung cancer (NSCLC) [5,11,13,38,46,47]. In metastatic colorectal cancer (mCRC), KRAS proto-oncogene, GTPase (KRAS) mutation is responsible for primary resistance to EGFR blockage [6,7,8,9]. Genotyping of tumors is recommended as routine practice in clinical oncology. While tissue biopsy is the gold-standard for genotyping, the feasibility of genotyping of ctDNAs in various kinds of cancer has been demonstrated [36,48,49,50].

An advantage of ctDNA testing compared with a tissue biopsy is that it is less invasive and allows sequential blood sampling. In a metastatic setting, ctDNAs are better diagnostic samples than single-site biopsied tissue since ctDNAs originate from multiple tumor sites [16,17,25,30,51]. For example, in the diagnosis of lung cancer, the lack of available tissues for molecular profiling, inaccessible tumor location, and the risk of complications in case of adverse events with image-guided biopsies are serious limitations for a tissue biopsy [52]. Delays often occur in tissue biopsy. In a prospective study of EGFR genotyping in advanced lung cancer, the median test turnaround time for tissue biopsy was 12 (1–53) days for a new diagnosis of non-squamous, non-small cell lung cancer. In contrast, the median time was 3 (1–7) days for ctDNA testing for the same [53]. ctDNA testing is cost effective as compared to tissue biopsy, which has the added risk of complications [52].

The alteration of ctDNAs reflect real time information which occurs in vivo. The half-life of ctDNAs in circulation has been estimated to be between 16 min and 2.5 h [19]. It has been reported that ctDNAs are ideal biomarkers to monitor response to therapy and emergence of secondary mutations associated with resistance to therapy, revealing heterogeneity and clonal evolution in cancer progression [12,16,25,26,29,39,40,45,51,54,55,56,57,58,59,60,61,62,63,64,65,66,67,68,69,70,71,72,73,74,75,76,77,78]. Thus, ctDNA testing is an appealing approach for the genotyping of individual tumors. Quantitative and molecular analysis of ctDNAs enables assessment of the dynamic changes like a ‘real-time’ snapshot of the disease.

## 3. Use of ctDNAs in Cancer Management

### 3.1. Lung Cancer

EGFR mutations are commonly observed in NSCLC and present in almost 50% of patients with advanced NSCLC [79,80]. Activating EGFR mutations are mainly the exon 19 deletions (Del19) and the L858R point mutation in exon 21, which are known to be the most important predictive factors for sensitivity to EGFR TKIs and are used for selection of EGFR TKIs in NSCLC [5,81]. First-generation TKIs, such as erlotinib and gefitinib, target the receptor via reversible binding of tyrosine kinase domain, while second-generation TKIs, such as afatinib, covalently bind the target [70]. Response and progression-free survival (PFS) with EGFR TKIs are superior to standard chemotherapy in NSCLC with activating EGFR mutations [39,46]. Detection of activating EGFR mutations for NSCLC by ctDNA testing shows high concordance with those by tissue biopsy, especially in specificity [39,40,49,53,60,65,82,83,84,85,86]. A meta-analysis examining 27 studies conducted between the years 2007 and 2015 demonstrated a pooled sensitivity of 0.62 (95% confidence intervals (CI), 0.51–0.72) and 0.96 (95% CI 0.93–0.98) for specificity in EGFR genotyping in NSCLC [49]. While tissue biopsy often provides limited and low-quality material for genotyping at the time of progression, monitoring of active EGFR mutation is described as a potential prognostic marker for the efficacy of EGFR TKIs [39,76,78,86]. High performance of characteristics of EGFR mutation by ctDNA testing was demonstrated in a real-world setting.

The substitution of threonine to methionine at amino acid position 790 (T790M) in exon 20 of EGFR gene reduces binding of first- and second-generation EGFR TKIs to the ATP-binding pocket of EGFR, thereby reducing response. T790M mutations account for approximately 50%–60% of the acquired resistance mechanisms [80]. Detection of T790M mutation by ctDNA testing has proved to be challenging due to low abundance in blood before the beginning of the treatment [87,88]. On the other hand, it has been reported that T790M mutations by ctDNA testing are observed in the course of treatment [59,66]. The third generation TKIs such as rociletinib (CO-1686, previously known as AVL-301) and osimertinib (previously known as AZD9291) target both activating EGFR mutations and T790M mutations [47,67,89,90,91]. Osimertinib was approved for patients with acquired T790M mutations, and the detection method for acquired T790M mutation includes both a tumor-tissue biopsy and ctDNA testing while testing the tumor tissue is the recommended method. Molecular analysis of ctDNAs allowed ongoing genomic analysis for patients on third-generation TKIs [62,70,92]. For instance, C797S mutation was identified as a novel key driver of resistance to osimertinib, while L798I mutation was reported in the resistance to rociletinib [62,70]. These studies showed that ctDNA testing could reveal clonal evolution and resistance to therapies, suggesting further implementation of ctDNA testing in clinical practice for lung cancer therapy in the near future.

### 3.2. Breast Cancer

Breast cancer is a heterogeneous disease [93]. Breast cancer is clinically categorized into three major subtypes, which show distinct characteristics and reflect patient prognosis: hormone receptor (HR)-positive type (oestrogen receptor [ER]+, progesterone receptor [PgR]+/−, and HER2-), HER2-positive type (ER-, PgR +/−, and HER2+), and triple-negative (TN) type (ER-, PgR-, and HER2-) [94]. Interestingly, it was reported that variations of somatic mutations across molecular subtypes are observed by molecular analysis of ctDNAs [95].

Endocrine therapy (ET) for breast cancer was the first ever targeted therapy used in any type of cancer. ET-based regimens form the backbone of the treatment for HR-positive type, while anti-HER2 treatment works for HER2-positive type [13,96]. ET includes a variety of agents like selective ER modulator (e.g. tamoxifen), selective ER down-regulator (e.g. fulvestrant), and aromatase inhibitors (AIs) (e.g. exemestane, letrozole, anastrozole). Today, combinations of additional drugs to ET have been developed to overcome resistance to acquired ET, for example, phosphatidylinositol 3-kinase (PI3K) inhibitors (e.g. buparlisib, alpelisib, taselisib), mammalian target of rapamycin (mTOR) inhibitors (e.g. everolimus, temsirolimus), and cyclin-dependent kinase 4 and 6 (CDK4/6) inhibitors (e.g. palbociclib, ribociclib, abemacilib) [97]. However, no specific biomarkers for the use of these new agents have been identified.

AIs are the current recommended first-line choice for ET [97]. One of the mechanisms of resistance to AIs is the activating mutations in the ligand-binding domain of ESR1, the gene coding for the ERα [98]. While ESR1 mutations rarely occur in primary breast cancer, enrichment of ESR1 mutations is observed in metastatic breast cancer (MBC) [98,99]. Notably, ESR1 mutations (e.g. Y537C/S/N, D538G) are frequently sub-clonal with high levels of polyclonality [100,101,102]. Several groups observed the existence of ESR1 mutations in MBC by ctDNA testing, suggesting that ctDNAs are a good substrate for detection of ESR1 mutations [99,101,102,103,104]. On the other hand, ESR1 epigenetic silencing potentially affects response to AIs. It was reported that ESR1 methylation in ctDNAs could be a potential biomarker for response to everolimus/exemestane treatment [105]. With regards to patient outcome, it was reported that ESR1 mutations in ctDNAs are associated with inferior outcomes [106,107,108]. Contrary to these studies, a phase III PALOMA-3 study for the assessment of palbociclib and fluvestrant efficacy reported that prediction of clinical outcome is limited by using ESR1 mutations in plasma samples [109,110].

Together with ESR1, PIK3CA, the p110 isoform of PI3K, is regarded as a promising biomarker. PIK3CA mutations are frequently observed in HR-positive MBC and are associated with activation of PI3K pathway [111]. In a phase III study that was the first randomized clinical trial involving PI3K inhibitors in MBC, detection of PIK3CA mutation by ctDNA testing showed improvement in PFS with buparlisib plus fluvestrant compared with fluvestrant alone (4.6 month vs. 1.5 month; hazard ratio (HR) 0.58, 95% CI 0.32–1.05, log-rank *p* = 0.036), while there was no significant difference in PFS between the PI3K pathway activated group and non-activated group identified by tissue sample sequencing [112]. While prognostic value of PIK3CA has not been elucidated in ET including new developing drugs, O’Leary et al. showed that PIK3CA ctDNA levels after 15 days’ treatment with palbociclib and fluvestrant strongly predicts PFS (HR 3.94, 95% CI 1.61–9.64, log-rank *p* = 0.0013) [109,113].

HER2 amplification is a critical biomarker conferring sensitivity in combination with anti-HER2 therapy [13]. It was demonstrated that the molecular analyses of ctDNAs could reveal the existence of amplified HER2 in ctDNAs [114]. However, sensitivity for HER2 detection in ctDNAs was relatively low [114]. On the other hand, it was reported that longitudinal gene-panel ctDNA sequencing could reveal the mechanism of resistance to pyrotinib, a TKI which has been developed for HER2-positive tumors [61]. In a phase II clinical trial that aimed to assess clinical benefits of neratinib, pan HER inhibitor in HER2-mutated non-amplified MBC, ctDNA HER2 mutant variant allele frequency was predictive of response to neratinib [115].

Unfortunately, effective targeted therapy for TN breast cancer has not been investigated yet. Majority of the TN type has mutations in breast cancer susceptibility gene (BRCA) 1/2. Response and PFS with olaparib, a poly adenosine diphosphate-ribose polymerase (PARP) inhibitor, is superior to standard chemotherapy in MBC with BRCA germline mutations (7.0 month vs. 4.2 month; HR 0.58, 95% CI 0.43–0.80, log-rank *p* < 0.001) [116]. One of the mechanisms of resistance to PARP inhibitor is from somatic reversion mutations or intragenic deletions that restore the functions of BRCA [117]. It was reported that BRCA1/2 reversion mutations could be detected by ctDNA sequencing analysis in patients with ovarian and breast cancer [118].

At present, most of the studies have failed to develop workable criteria of ctDNA testing for clinical practice in MBC [61,119]. However, molecular analysis of ctDNAs is an appealing alternative approach for the characterization of tumor molecular heterogeneity and its evolving biology [22,26]. Thus, ctDNA testing may provide important clues to investigate dedicated predictive biomarkers for new drugs since a wide range of agents are being developed for MBC.

### 3.3. Colorectal Cancer

Monoclonal EGFR antibodies such as cetuximab and panitumumab are standard agents of treatment regimens for mCRC, either alone or in combination with chemotherapy. Addition of EGFR antibodies has improved patient survival [6,7,8,9]. In clinical practice, the identification of RAS mutations is required before initiating treatment since RAS mutations are regarded as critical biomarkers of innate resistance to EGFR inhibitors [6]. Currently, determination of RAS mutation status is performed using formalin-fixed paraffin-embedded tumor tissues. Molecular analysis of ctDNAs can be used as an alternative to tissue analysis. A meta-analysis examining 31 studies conducted between the years 2000 and 2017 demonstrated a pooled sensitivity of 0.64 (95% CI, 0.61–0.67) and 0.94 (95% CI 0.93–0.96) for specificity in RAS mutations in CRC [50]. Previous studies have demonstrated RAS mutations by ctDNA testing as an early marker of therapeutic response [34,45,120]. In addition, the emergence and the progressive increase of detectable RAS mutations prior to subsequent progression by ctDNA testing have been demonstrated [25,45,54,71,72,73,74].

A treatment strategy for patients who respond and then relapse due to resistance to EGFR inhibitors is urgently required. Most frequent secondary mutations occur in KRAS and NRAS, which are presently untreatable as the corresponding proteins are fractious to pharmacological blockage [121]. There are only very few available treatment strategies based on molecular rationale in mCRC after failure of EGFR blockage. HER2 amplification is an emerging biomarker in colorectal cancer that confers to combination anti-HER2 treatment and predicts resistance to EGFR blockage, although the frequency of HER2 amplification is relatively low. It was reported that clinically validated ctDNA testing could be a reliable diagnostic of HER2 copy number in plasma that predicted response rates to trastuzumab and lapatinib in mCRC [122]. Upon failure of chemotherapy plus EGFR antibodies, CRC patients usually stop additional EGFR antibodies, while re-challenge of EGFR antibodies could provide clinical benefits in molecularly selected patients beyond second line [123]. Interestingly, Parseghian et al. demonstrated clinical benefits of re-challenge of EGFR blockage by capturing the mutant minimal drop of RAS levels in blood and reinitiating treatment [77].

Recently, Russo et al. reported that the profile of the LMNA-NTRK1 rearrangement in ctDNAs paralleled tumor response and resistance to entrectinib (RXDX-101, previously known as NMS-E628), a potent pan tropomyosin-related kinase (TRK), ALK, and ROS1 inhibitor [124]. It was demonstrated that molecular analysis of ctDNAs in CRC patients could provide new information of mutation status during the course of treatment and reveal resistance mechanisms [20,25,45,54,71,72,73,74]. These studies suggested the usefulness of application of ctDNAs for guiding treatment decision in CRC [20,34,54,71,74,77,125].

## 4. Conclusions and Future Perspectives

In oncology, ctDNAs are promising biomarkers to guide clinical decision-making. Several clinical studies have been designed to further explore the utility and feasibility of this approach. Currently, there is an unmet need for predictive biomarkers of response to immune check point inhibitors such as the programmed death ligand 1 (PD-L1) inhibitors, programmed death 1 (PD1) inhibitors, and CTLA-4 antibody. High alterations in cfDNAs were related to the favorable outcomes with checkpoint inhibitor-based immunotherapy across various histologies [126]. For instance, tumor mutational burden (TMB) from cfDNAs was reported as a predictive biomarker for PFS in patients receiving atezolizumab (an anti PD-L1) in NSCLC [127]. ctDNA testing is used to accurately and reproducibly measure TMB, suggesting that ctDNAs can be a predictive biomarker in deciding the adaptation of immunotherapy. Although we focus on the use of ctDNAs in cancers with advanced stages in this review, the utility and feasibility of ctDNAs have been demonstrated in cancers at early stages as well. Further applications of ctDNAs in clinical practice require optimization, standardization, and validation of measuring ctDNAs for each purpose. Broadening our knowledge of ctDNAs, including prior knowledge of actual kinetics, will offer opportunities for non-invasive cancer management that opens new avenues for clinical practice in the near future.

## Figures and Tables

**Figure 1 jcm-08-01365-f001:**
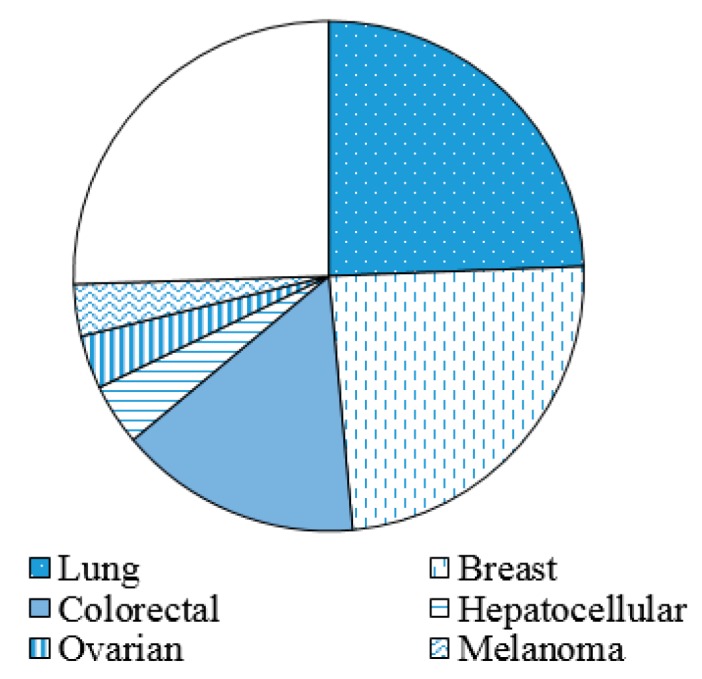
**Proportion of the publication numbers reporting the use of circulating DNA in cancer types**. This is a modification of Figure 4 in Reference [21], web of science citation reports showing 5800 records for the cancer circulating DNA up to the end of 2018. The most studied cancer subtypes are lung cancer, breast cancer, and colorectal cancer, in that order.

**Figure 2 jcm-08-01365-f002:**
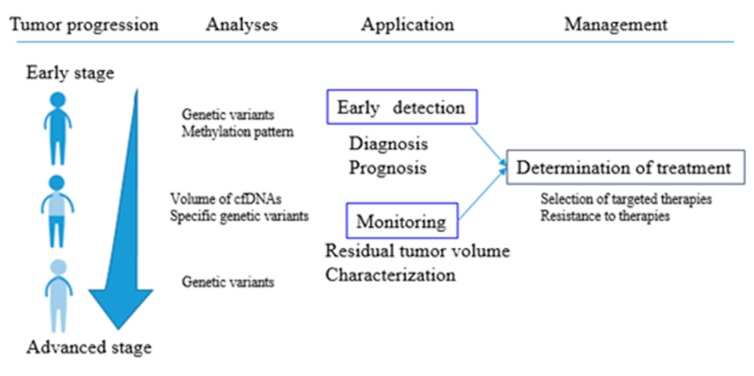
**ctDNAs as promising biomarkers in the different phases of cancer progression**. The schematic summarizes the most suitable clinical applications in each phase. In early stage, genetic variants and methylation patterns of ctDNAs might be useful for early diagnosis. Relevant prognostic information can be provided by analysis of genetic variants of ctDNAs. In particular, specific genetic alterations and the volume of ctDNAs can be used for the detection of the minimal residual disease. In metastatic setting, analysis of ctDNAs might be an alternative to tissue analysis for the identification of predictive biomarkers for the therapeutic adaptation/response. This schema suggests its clinical application to guide decision-making in cancer treatment.

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
