# Peer review of "Towards Circulating-Tumor DNA-Based Precision Medicine"

_jcm, 2019, doi:10.3390/jcm8091365_

Round 1

Reviewer 1 Report

Ai Hironaka-Mitsuhashi et al address a very interesting topic in the molecular diagnostics landscape, but this review could be of greater relevance if it was integrated deepening some fundamental aspects of the cfDNA analysis. Indeed, the manuscript is missing from a part that deals with the theme of the “Approaches to cfDNA analysis”. In particular, it is fundamental to discuss about the pre-analytical and analytical phase because are very stringent aspect in the quality of the sample to be analyzed, especially if we consider the cfDNA a molecular biomarker to cancer monitoring. I believe that Figure 1 is not necessary and that Figure 2 should be better conceived, because the proposed scheme is not easy to read.

Author Response

We are grateful to the reviewers for their critical comments and insightful suggestions that have helped us considerably to improve our paper. We have taken all of these comments and suggestions into consideration in the revised version of our manuscript.

Response to the comments from Reviewer #1

Q1. The manuscript is missing from a part that deals with the theme of the “Approaches to cfDNA analysis”. In particular, it is fundamental to discuss about the pre-analytical and analytical phase especially if we consider the cfDNA a molecular biomarker to cancer monitoring.

A1. Thank you very much for your comment. We also agree that it is fundamental to discuss the pre-analytical and analytical phase for clinical application of molecular biomarkers. Since ctDNA testing was already approved as a companion diagnostic for lung cancer as mentioned in Introduction (line 64-66, page 2) and the part of lung cancer (line 147-149, page 4) we believe that the discussion for the pre-analytical and analytical phase of cfDNAs, “Approaches to cfDNA analysis” would be outside the scope of our manuscript. Thus, we added these sentences in text (line 66-67, page2) as followed;

For instance, EGFR (epidermal growth factor receptor) mutation testing using ctDNAs was approved as a companion in vitro diagnostic while ctDNA testing has required for the pre-analytical and analytical phase in the other cancer. 

Q2. Figure 1 is not necessary, and Figure 2 should be better conceived because the proposed scheme is not easy to read.

A2. Thank you very much for your comments and valuable suggestion. We simply made Figure 1 to clarify why we decided to focus on lung cancer, breast cancer, and colorectal cancer among various other types of cancers. Figure 2 shows that the purpose of the use of cfDNA as a reliable biomarker is different from the progression of cancer, and it is considered as a necessary figure for Introduction. As the reviewer requested, we have modified Figure 2. We added underlining to separate factors and descriptions. When we checked the figure again, we realized that there was no explanation of ‘early detection’, and ‘monitoring’, so we added ‘application’ as factor in the figure. We also changed the terms in Figure 2, ‘analysis of cfDNAs’ and ‘progression’ as followed; ‘analyses’ and ‘tumor progression’.

Reviewer 2 Report

this is a well written nicely presented review of the literature regarding the theories of how ctDNA occurs, the techniques for detection as well as the evidence of its use in lung, breast and colorectal cancers. 

Figures are well presented and add to the manuscript. 

There are some minor corrections to inflows and grammar needed but overall I would recommend publication of this review with minor corrections. 

Line 37- improvement ‘in’ (not ‘to’)

Line 41- fluids, collectively termed liquid biopsies.

I support the publication of this article with very minimal corrections, as above. 

Author Response

We are grateful to the reviewers for their critical comments and insightful suggestions that have helped us considerably to improve our paper. We have taken all of these comments and suggestions into consideration in the revised version of our manuscript.

Response to the comments from Reviewer #2

Q1. There are some minor corrections to inflows and grammar needed.

(a) Line 37- improvement ‘in’ not ‘to’.

(b) Line 41- fluids, collectively termed liquid biopsies.

A1. We have been honored by your comments. Thank you very much. We have edited our manuscript accordingly.